# Short communication: Evaluation of charged membrane filters and buffers for concentration and recovery of infectious salmon anaemia virus in seawater

Simon Chioma Weli👤*, Haitham Tartor, Bjørn Spilsberg, Ole Bendik Dale, Atle Lillehaug

Norwegian Veterinary Institute, Oslo, Norway

* Simon.Weli@vetinst.no

## Abstract

Infectious salmon anaemia virus (ISAV) is the cause of an important waterborne disease of farmed Atlantic salmon. Detection of virus in water samples may constitute an alternative method to sacrificing fish for surveillance of fish populations for the presence of ISA-virus. We aimed to evaluate different membrane filters and buffers for concentration and recovery of ISAV in seawater, prior to molecular detection. One litre each of artificial and natural seawater was spiked with ISAV, followed by concentration with different filters and subsequent elution with different buffers. The negatively charged MF hydrophilic membrane filter, combined with NucliSENS® lysis buffer, presented the highest ISAV recovery percentages with 12.5 ± 1.3% by RT-qPCR and 31.7 ± 10.7% by RT-ddPCR. For the positively charged 1 MDS Zeta Plus® Virosorb® membrane filter, combined with NucliSENS® lysis buffer, the ISAV recovery percentages were 3.4 ± 0.1% by RT-qPCR and 10.8 ± 14.2% by RT-ddPCR. The limits of quantification (LOQ) were estimated to be $2.2 \times 10^3$ ISAV copies/L of natural seawater for both RT-qPCR and RT-ddPCR. The ISAV concentration method was more efficient in natural seawater.

## Introduction

Infectious salmon anaemia virus (ISAV) is a negative sense single-stranded RNA virus that belongs to the family *Orthomyxoviridae*, genus *Isavirus* [1–4]. ISAV is the causative agent of infectious salmon anaemia (ISA), which is arguably one of the most important waterborne viral disease of farmed Atlantic salmon [5–8]. The OIE-listed disease is characterized by anaemia and circulatory disturbances, such as oedemas, ascites and bleedings, and can result in up to 90% mortality in affected fish populations [9]. The mechanisms of ISAV transmission are not fully understood; however, the majority of ISA disease outbreaks occur in the seawater production phase, and shedding of virus from the mucus and faeces of infected fish into seawater has been suggested as a major source of spread [6].

ISA disease outbreaks caused by horizontal transfer of the virulent virus have been reported in several studies, indicating possibilities of virus spread between farms [10–13]. Therefore, routine surveillance of seawater for the presence of ISAV can be a vital supplement to current

**Data Availability Statement:** All relevant data are within the paper and S1 Fig, S1 Table and S1 Data.

**Funding:** This research work was funded by grants from the Fiskeri - Og Havbruksnæringens Forskningsfond, 267411, and the Norwegian Fishery Ministry grant: Biosecurity in Aquaculture (13055).

**Competing interests:** The authors have no competing interests.

surveillance and control methods based on detection of clinical ISA disease, thereby capturing information on the early presence of the virus. Current laboratory methods used for diagnosis of ISA and detection of ISAV are largely dependent on fish sampling for nucleic acid isolation and amplification from tissue samples, and histopathology and immunohistochemistry analysis [8]. Fish sampling, in general, is selective, resource-demanding, and may be problematic from an animal welfare point of view. These problems could be circumvented by the development of methods that can be used to concentrate and detect ISAV from seawater instead of from fish.

Several studies have shown the possibilities of using different filtration approaches to concentrate viruses and other infectious agents from tap water, groundwater, fresh- and seawater [14–18]. The principle of eDNA detection, which involves the detection of genetic material from various organisms in water samples, in order to identify the presence of the actual organisms in the water bodies (lakes, rivers), has also been applied for surveillance purposes of disease agents in wild aquatic animals in freshwater, like crayfish plague (*Aphanomyces astaci*) in Noble crayfish (*Astacus astacus*) [19] and *Gyrodactylus salaris* in Atlantic salmon (*Salmo salar*) [20]. Moreover, we have promising results regarding concentration and detection of *Salmonid alphavirus* from seawater both in laboratory experiments [21] and infection trials with Atlantic salmon [22].

Since salts and other RT-PCR inhibitors in environmental water can have influence on virus detection and quantification [23], reverse transcriptase droplet digital PCR (RT-dd-PCR) has been suggested as an alternative method due to the principle of serial dilution of the target molecules. Recent studies have shown an improved detection and quantification of RNA viruses from water samples with high salt and inhibitory contents using RT-ddPCR [23–29].

In this study, we aim to evaluate the concentration and detection method developed for Salmonid alphavirus for use in concentrating ISAV from seawater. The experiments involved spiking of ISAV in natural and artificial seawater and concentration by adsorption to and elution from membrane charged filters, followed by virus detection with reverse transcriptase droplet digital PCR and quantitative RT-qPCR.

## Materials and methods

### Virus and cell cultures

The ISAV-Glesvær 2/90 isolate was propagated in an Atlantic salmon kidney (ASK) cell line, as previously described [30, 31]. Briefly, cells were grown at 20˚C in T-150 culture flasks containing Leibowitz L-15 medium (Life Technologies, UK), supplemented with 10% foetal calf serum (FBS) and gentamicin (Lonza, USA). The ISAV isolate was used to inoculate the cell culture, and after seven-day incubation at 15˚C, the culture supernatants were harvested, centrifuged, and 1 mL aliquots were prepared and stored at -80˚C prior to be used for spiking of the water.

### Water samples and virus spiking

In this study, two types of seawater were used for virus spiking. (1) The artificial seawater was obtained from Sigma-Aldrich, Germany, in 1 L polyethylene bottles, and was kept at 4˚C in the dark before use. (2) The natural seawater was collected from the Oslo-fjord, and was kindly provided by the Norwegian Institute for Water Research (NIVA), Solbergstrand research station. For virus spiking, 1L of seawater sample was spiked with $3.5 \times 10^7$ ISAV copies.

### ISAV concentration procedures

The same filters, buffers and concentration procedures described previously [21] were used to concentrate ISAV from 1L of artificial seawater or natural seawater spiked with ISAV ($3.5 \times 10^7$

copies), using either MF- or MD+ filters. Five different concentration methods, with different combinations of filters and buffers were used:

- MF$^-$/buffer 1: NucliSENS® lysis buffer (bioMérieux SA, France)

- MD$^+$/buffer 1: NucliSENS® lysis buffer (bioMérieux SA, France)

- MF$^-$/buffer 2: 1 mM NaOH pH 9.5 buffer

- MF$^-$/buffer 3: Leibowitz L-15 medium (Life Technologies, UK) + 2% FBS (pH 9.0)

- MD$^+$/buffer 4: Leibowitz L-15 medium (Life Technologies, UK) + 2% FBS (pH 7.5)

To elute the adsorbed virus particles from the membrane filters, four different buffers were tested. For the MF$^-$/buffer 1 method, the filter was immediately placed into a Petri dish containing 2.4 mL of the buffer and shaken for 30 min at 600 RPM, in room temperature. For elution with buffer 2, 3 and 4 methods, each filter was cut into small pieces and placed in a 50 mL Falcon tube containing 4mL of the respective buffer. Each sample was vortexed (3 x 1 min, with 5 min intervals at room temperature), and concentrates were stored at -80˚C prior to RNA extraction and ISAV detection by RT-qPCR and RT-ddPCR (Fig 1). For both types of

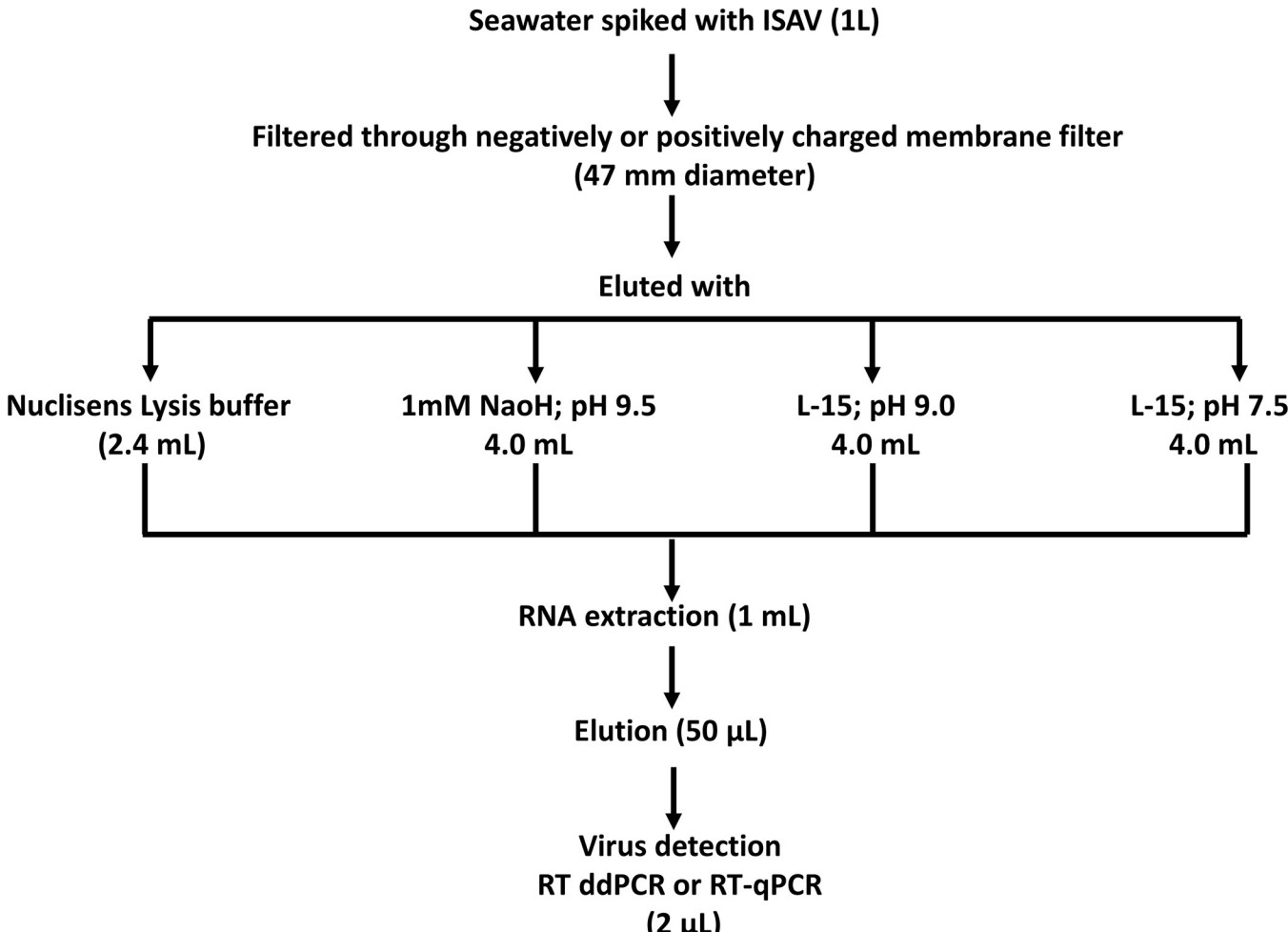

**Fig 1. Schematic illustration of different approaches of ISAV concentration from seawater samples and virus detection, using RT-qPCR and RT-ddPCR methods.**

seawater, a negative control sample without ISAV spiking was analysed to verify the absence of any natural contaminants. Retrospective (back) calculations were used to calculate ISAV copy numbers in 4 or 2.4 ml concentrates by taking into consideration, that ISAV was spiked in 1L of seawater, concentrated and eluted in either 4 or 2.4 ml (lysate) and that from the 4 or 2.4ml, RNA was extracted from 1ml of the lysate. The extracted RNA was eluted in 50 μL and 2 μL of RNA was used for PCR. The ISAV recovery percentages in the concentrates obtained using different filters and buffers were calculated based on the formula:

$$\text{ISAV recovery } (\%) : \frac{\textit{ISAV copy number in seawater concentrate}}{\textit{ISAV copy number in spiked seawater}} \, x \, 100$$

## RNA extraction

Total RNA was extracted from 1 mL from all samples assigned for RNA extraction according to the generic easyMAG protocol (bioMèrieux, Marcy l'Etoile, France). The RNA was eluted in 50 μL of elution buffer and stored at −80˚C prior to PCR analysis.

## RT-qPCR-setup

The RT-qPCR was carried out to detect and quantify ISAV using the AriaMx Realtime PCR system (Agilent Genomics, USA). The ISAV-M-segment 8-specific primers (F-primer: 5'-CTACACAGCAGGATGCAGATGT-3', R-primer: 5'-CAGGATGCCGGAAGTCGAT-3') and probe (5'-6FAM-CATCGTCGCTGCAGTTC-MGBNFQ-3') were used to generate a 105-bp product [32], using TaqMan® Fast Virus 1-Step Master Mix kit (Applied Biosystems, USA). Two μl of isolated RNA was used in a total volume of 20μL RT-qPCR mix containing 500 nM of each primer, 300 nM probe and two parallels were analysed for each sample. The following PCR conditions were used: reverse transcriptase at 50˚C for 5 min, initial denaturation at 95˚C for 2 min, and 45 cycles of amplification (94˚C for 15 s, 60˚C for 40 s). Environmental samples, such as seawater are particularly challenging and usually contain complex mixtures of a variety of inhibitory substances that could impact the RT-qPCR and target quantification. For this reason, ISAV RNA was, analyzed undiluted (1:1) and diluted (1:4) in duplicates, by RT-qPCR. Potential inhibition was detected when the Cq difference between the 1:1 and 1:4 samples was found to be less than 2 Cq. For the samples that fall within this threshold, the 1:4 dilution was used to estimate virus quantities. The ratio 1:1 indicate stock RNA (2 μL template) while the 1:4 ratios was stock RNA: RNA elution buffer (i.e. 1ul RNA + 3 μL RNA elution buffer = 1:4).

## Reverse Transcriptase digital droplet PCR (RT-ddPCR)

The primers and probe used for the RT-qPCR were also used for the RT-ddPCR assay, by optimizing annealing temperature and primer/probe concentrations. Three different primer/probe concentrations were tested: (1) low; with 600 nM primers and 150 nM probe, (2) medium; 900 nM primers and 250 nM probe, and (3) high; with 1200 nM primers and 350 nM probe. The medium concentration is the recommended starting point from the instrument vendor (Bio-rad). These three conditions were run with 8 different annealing temperatures in a gradient PCR with a constant amount of the relevant template on a Q X200 system (Bio-Rad). A one-step RT-ddPCR advanced kit mastermix (Bio-Rad) was used, and PCR was performed in a C1000 Touch thermal cycler (Bio-rad) with a 96- deep well reaction module. The temperature profile for PCR was: 60˚C for 60 min, 95˚C for 10 min, followed by 45 cycles of 95˚C for 30 s, from 50.0 to 63.0˚C annealing for 1 min, and a final 98˚C for 10 min. For all steps a ramp rate of 2.5˚C/min was used. Results were analysed with Quantasoft 1.7.4.0917

software (Bio-Rad). The threshold for distinguishing positive from negative droplets was set manually.

## Quantification of ISAV stock

The RT-ddPCR was used to quantify ISAV in stock samples, by measuring the absolute copy number of the virus without a need for a reference standard. For this purpose, two-fold serial dilutions (1:1 to 1:$2^9$) of the ISAV stock sample were prepared, the total RNA was extracted from each dilution and the absolute ISAV copy numbers in 2 μL RNA of all dilutions were measured by RT-ddPCR. In parallel, all the stock diluted samples (1:1 to 1:$2^9$) were also analysed for their ISAV *Cq* values using RT-qPCR. The absolute ISAV copy numbers (measured by RT-ddPCR) and *Cq* values (measured by RT-qPCR) from the undiluted stock sample were used to calculate the ISAV copy numbers in the stock dilutions (1:1 to 1:$2^9$) based on their *Cq* values, using the formula: *N1 = N2\*(1+E)$^{(CqN2-CqN1)}$* [33], where N1 and N2 denote ISAV copy numbers in the diluted and undiluted stock samples, respectively, *E* is the amplification efficiency of the ISAV RT-qPCR (109% = 1.09), and *Cq* is the ISAV cycle quantification values.

## Efficiency of the MF⁻/buffer 1 method for concentration of ISAV

The efficiency of the MF⁻/buffer 1 method to concentrate ISAV in seawater was assessed using different concentrations of ISAV to spike 1 L natural seawater as described previously [21]. After concentration, the total RNA was isolated from the stock sample, stock dilutions and the concentrated samples, and ISAV was detected and quantified using RT-qPCR and RT-ddPCR (direct measurement). For calculation of ISAV copy number based on the *Cq* values (RT-qPCR), we used the formula *N1 = N2\*(1+E)$^{(CqN2-CqN1)}$*, where N1 and N2 denote the ISAV copy numbers in the diluted/concentrated samples and the stock sample, respectively, *E* is the amplification efficiency of the ISAV RT-qPCR (109% = 1.09), and *Cq* is the ISAV cycle quantification values. The recovery percentages were calculated as shown before. The whole experiment was performed in quadruplicate, and the average of the four ISAV recovery percentages at each virus concentration was calculated.

## Limit of quantification (LOQ) and sensitivity of ISAV in 1 L natural seawater

Estimation of the sensitivity and LOQ of the MF⁻/buffer 1 concentration method of ISAV from 1 L seawater sample was determined by analysing the virus quantity recovered (quantified by RT-qPCR and RT-ddPCR) from the concentrated samples after virus spiking. For this purpose, eleven five-fold serial dilutions (1: 1 to 1: $5^{11}$) were prepared from ISAV stock (3.5 x $10^7$ copies/mL) as described previously [21]. After concentration, total RNA was extracted from stock dilutions and the concentrated samples, and ISAV was detected and quantified by RT-qPCR and RT-ddPCR, as explained before. The experiment was performed in duplicates, and the average of virus recovery in the two experiments was calculated. The lowest quantity of ISAV in the spiked seawater sample that could be concentrated by MF⁻ /buffer 1 method and detected using the PCR methods was taken as the sensitivity, while the LOQ was calculated from the highest dilution giving results with an RSD of < 25% [34].

## Statistical analysis

All statistical analyses were performed using Statistical Data Analysis Software JMP®. A student *t-test* was used to compare differences in ISAV recovery between the RT-ddPCR and RT-

qPCR assays from two-fold serial dilutions (1:1 to 1:2$^9$). A difference of $p < 0.05$ was considered statistically significant.

# Results

## Optimization of RT-ddPCR conditions

Well-separated clusters of positive and negative droplets are a prerequisite for correct determination of concentrations using RT-ddPCR. We optimized the RT-ddPCR conditions for detection of ISAV by testing different primer/probe concentrations versus different annealing temperatures Table 1. The absolute value of fluorescence increased for both positive droplets and negative droplets increased with increasing primer and probe concentrations (Fig 1), Table 1 and S1 Fig. However, the ratio between positive and negative droplet fluorescence remained relatively constant with values from 1.9 to 2.1. Taken together, all the tested conditions appeared to be acceptable and indicating that the assay is quite robust. An annealing temperature of 60˚C was chosen, together with 900 nM primer concentration and 250 nM probe concentration as conditions for detection of ISAV with RT-ddPCR in the following experiments.

## ISAV quantification in virus stock samples

Two different quantitative PCR platforms (RT-qPCR and RT-ddPCR) were used to quantify ISAV in the virus stock sample. Both platforms exhibited similar PCR efficiencies that resulted in the quantification of ISAV in the range between 3.5 x 10$^7$ and 3.3 x 10$^4$ copies/mL (RT-qPCR) and 3.5 x 10$^7$ and 4.1 x 10$^4$ copies/mL (RT-ddPCR) at dilution range between 1:1 and 1:2$^9$, respectively. The ISAV quantification in the two-fold serial dilutions of ISAV the stock sample, using RT-ddPCR, showed a corresponding two-fold reduction in the virus quantity (copy/mL) (Fig 3). Quantitative agreements were found between ISAV copies obtained from both RT-ddPCR and RT-qPCR over seven different dilutions (1:1 to 1:2$^8$), with a noticeable deviation observed only at the 1:2$^9$ dilution for the RT-ddPCR.

## Evaluation of filters and buffers for virus recovery

In this study, different filters and buffers were evaluated for their capacity to concentrate ISAV in artificial and natural seawater. The results showed that the percentage of ISAV recovered from natural seawater was, in general higher than those concentrated from the artificial seawater, as determined by both the RT-ddPCR and RT-qPCR data Table 2. Regardless of the type of seawater analysed, the MF$^-$ filter gave higher ISAV recovery as compared to the MD$^+$ filter

**Table 1. Shows ISAV copy numbers measured with RT-ddPCR at different annealing temperatures and primer/probe concentrations.**

| Temperature | ISAV quantity (copies/µL) measured at different primer/probe concentrations (nM) | | |
|---|---|---|---|
| | Low (600/150) | Medium (Recommended; 900/250) | High (1200/350) |
| 63.0 | 1722 | 2167 | 1867 |
| 62.4 | 1722 | 2167 | 1922 |
| 61.4 | 1633 | 2089 | 1978 |
| 59.9 | 1833 | 2333 | 2100 |
| 58.1 | 1867 | 2267 | 2133 |
| 56.5 | 1967 | 2533 | 2200 |
| 55.6 | 2000 | 2467 | 2078 |
| 55.0 | 2056 | 2444 | 2144 |
| Average | 1850 | 2308.4 | 2052.8 |

**Table 2. Shows the percentages of ISAV recovery after concentration of the virus from artificial and natural seawater using different approaches.** One litre of seawater was spiked with $1.62 \times 10^7$ copies of ISAV before the virus was concentrated using a negatively charged (MF⁻) or a positively charged (MD⁺) filter. Four different buffers were compared for their elution capacity of the virus absorbed on the filters. Recovery rate (%) of ISAV in spiked artificial and natural seawater was quantified by RT-qPCR and RT-ddPCR. ISAV recovery is presented as mean ± standard deviation from two biological replicates.

| Membrane | Sample type | Elution buffer | ISAV recovery (%) | |
|---|---|---|---|---|
| | | | RT-qPCR | RT-ddPCR |
| MF negatively charged filter | Artificial seawater | NucliSENS® buffer | 9.5 ± 4.7 | 14.2 ± 14.1 |
| | | 1m M NaOH pH 9.0 | 0.1 ± 0.0 | 0.6 ± 0.4 |
| | | L-15 + 2% FBS pH 9.0 | 0.4 ± 0.0 | 1.9 ± 1.4 |
| | | L-15 + 2% FBS | 0.3 ± 0.0 | 1.5 ± 1.0 |
| | Natural seawater | NucliSENS® buffer | 12.5 ± 1.3 | 31.7 ± 10.7 |
| | | 1m M NaOH pH 9.0 | 0.6 ± 0.2 | 4.7 ± 3.8 |
| | | L-15 + 2% FBS pH 9.0 | 0.3 ± 0.2 | 3.5 ± 3.7 |
| | | L-15 + 2% FBS | 0.2 ± 0.0 | 3.4 ± 3.3 |
| MD positively charged filter | Artificial seawater | NucliSENS® buffer | 2.1 ± 0.6 | 5.8 ± 7.4 |
| | | 1m M NaOH pH 9.0 | ND | ND |
| | | L-15 + 2% FBS pH 9.0 | 0.1 ± 0.1 | 0.7 ± 0.1 |
| | | L-15 + 2% FBS | ND | 0.2 ± 0.2 |
| | Natural seawater | NucliSENS® buffer | 3.4 ± 0.1 | 10.8 ± 14.2 |
| | | 1m M NaOH pH 9.0 | ND | 0.1 ± 0.0 |
| | | L-15 + 2% FBS pH 9.0 | ND | 0.1 ± 0.1 |
| | | L-15 + 2% FBS | ND | ND |

ND = Not detected.

Table 2. Reverse transcriptase-ddPCR results showed the MF⁻/buffer 1 and MD⁺/buffer 1 methods produced the highest ISAV recoveries from natural seawater, with 31.7 ± 10.7% and 10.8 ± 14.2% (mean ± standard deviation), respectively, compared to that obtained using each of the filters in combination with buffer 2, 3 and 4 Table 2.

## Efficiency of MF⁻ /buffer 1 method in concentrations of ISAV in 1 L seawater

As the initial experiment showed that the MF⁻/buffer 1 method had the highest ISAV recovery percentages, as compared to the other methods Table 2, the concentration efficiency of this method was further evaluated by using different ISAV concentrations to spike 1 L seawater. Our results showed that ISAV recovery based on the RT-ddPCR quantification data had a minimum percentage of 31.3 ± 7.6 at 1:1 diluted stock and a maximum percentage of 84.1 ± 30.2 at 1:256 diluted stock Table 3. With few exceptions, low ISAV quantities in the spiked water led to high virus recovery percentage in the concentrated samples, and the concentration efficiency of the MF⁻/buffer 1 method seems to be, in general, higher when lower a concentration of virus is used for spiking Table 3. The quantification of ISAV in the concentrated samples, using RT-qPCR, showed a lowest virus recovery percentage of 15.4 ± 3.7 at 1:1 diluted stock and a highest percentage of 41.6 ± 12.6 and 41.6 ± 4.3 at 1:64 and 1:512, respectively. With few exceptions, the virus recovery calculated based on RT-qPCR quantification data showed significant lower percentages than that calculated based on RT-ddPCR data at different virus spiking concentrations Table 3, suggesting a better performance of the latter in the detection of ISAV after the concentration procedure. The linear regression analysis was applied to the transformed ($\log_{10}$) ISAV copy numbers/mL determined by RT-qPCR and RT-ddPCR, to compare virus quantities between samples before and after concentration using the

**Table 3. Show the difference in ISAV recovery percentages at different virus spiking concentrations for RT-qPCR and RT-ddPCR.** Two-fold serial dilutions of ISAV stock sample were used to spike different 1L natural seawater (1 dilution/each), and the spiked seawater samples were concentrated using MF⁻/buffer 1 concentration method. ISAV recovery percentages (%) and virus copy numbers/L are presented as mean ± standard deviation from four biological replicates. Student t-test was used to compare differences in ISAV recovery between RT-ddPCR and RT-qPCR assays.

| RT-qPCR | | RT-ddPCR | |
|---|---|---|---|
| ISAV dilutions (copy number/L) | % ISAV recovery in concentrated samples | ISAV dilutions (copy number/L) | % ISAV recovery in concentrated samples |
| 1:1 (35583474.2) | 15.4 ± 3.7** | 1:1 (34927777) | 31.3 ± 7.6** |
| 1:2$^1$ (14179312.0) | 22.9 ± 10.2 | 1:2$^1$ (1634722) | 35.7 ± 18.2 |
| 1:2$^2$ (7292671.3) | 24.8 ± 8.5 | 1:2$^2$ (7992361) | 48.9 ± 23.9 |
| 1:2$^3$ (2881041.5) | 26.9 ± 6.9* | 1:2$^3$ (3745833) | 54.8 ± 21.3* |
| 1:2$^4$ (1992519.7) | 23.4 ± 3.6* | 1:2$^4$ (2425000) | 41.1 ± 15.3* |
| 1:2$^5$ (713983.5) | 37.9 ± 2.4* | 1:2$^5$ (1049444) | 50.7 ± 12.4* |
| 1:2$^6$ (294636.4) | 41.6 ± 4.3* | 1:2$^6$ (407500) | 57.3 ± 16.2* |
| 1:2$^7$ (210392.5) | 22.2 ± 4.7* | 1:2$^7$ (218333) | 56.6 ± 19.7* |
| 1:2$^8$ (74670.8) | 31.6 ± 7.0* | 1:2$^8$ (78055) | 84.1 ± 30.2* |
| 1:2$^9$ (33434.2) | 41.6 ± 12.6* | 1:2$^9$ (41597) | 79.7 ± 28.1* |

Note

*$p < 0.01$

***$p < 0.001$ signs represent the levels of significance.

MF⁻/buffer 1 method. The results showed quantitative agreements of ISAV between the 2-fold serial dilutions of ISAV stock and corresponding concentrated samples, with coefficients of correlation, $R^2$ values, of 0.986 and 0.997 for the RT-qPCR or the RT-qPCR assay, respectively, indicating good linearity (Fig 4).

## Sensitivity and LOQ of ISAV in natural seawater concentrated by the MF⁻/buffer 1 method

The sensitivity and LOQ were estimated for ISAV by spiking 1 L natural seawater, and then concentrated with the MF⁻/buffer 1 method, before analysis with RT-qPCR and RT-ddPCR. The RNA extracted from the spiked natural seawater concentrate was analysed by RT-qPCR and RT-ddPCR to the lowest amount of ISAV RNA. A sensitivity of 2.2 x 10$^3$ copies/L and 4.4 x 10$^2$ were estimated based on the quantification by RT-qPCR and RT-ddPCR, respectively Table 4. LOQ of 2.2 x 10$^3$ copies/L for both RT-qPCR and RT-ddPCR were estimated for ISAV concentrated in the spiked 1 L natural seawater Table 4, which is defined as the limit of quantification of ISAV with RSD < 25% [34].

## Discussion

This study investigated the use of different filters and buffers for concentration and detection of ISAV in seawater, using qPCR assays. The results from the experiments showed that ISAV can be detected in artificial and natural seawater spiked with ISAV, as demonstrated by the concentration of water samples by both negative and positive charged filters. The negatively charged MF⁻ filter resulted in higher virus recovery percentages, as compared to the positively charged MD⁺ filter, while the NucliSENS® lysis buffer (buffer 1) was best when used to elute virus from the filters. Thirty-one percent of ISAV was recovered from natural seawater with the MF⁻/buffer 1 method by RT-ddPCR, clearly demonstrating that the MF⁻/buffer 1 method has the potential to be used for ISAV recovery under field conditions with farmed salmonids. The results shown here, share similarities with our previous study that showed the

**Table 4. Estimation of lower limit of quantification of ISAV in natural seawater.** Five-fold serial dilutions of ISAV stock were used to spike different 1L natural seawater samples (1 dilution/each) and the spiked seawater samples were concentrated using the MF-/buffer 1 concentration method. ISAV recovery rate (%) based on RT-qPCR and RT-ddPCR quantification data are presented as mean ± standard deviation from two biological replicates.

| ISAV dilutions | Dilution corresponding ISAV copy number | % ISAV recovery | |
|---|---|---|---|
| | | RT-qPCR | RT-ddPCR |
| 1:1 | 34927777 | 18.9 ± 1.9 | 30.5 ± 11.9 |
| 1:5 | 6985555 | 26.7 ± 8.0 | 30.4 ± 11.9 |
| $1:5^2$ | 1340014 | 23.9 ± 5.9 | 29.5 ± 16.5 |
| $1:5^3$ | 279422 | 21.5 ± 7.2 | 28.2 ± 13.4 |
| $1:5^4$ | 55884 | 16.5 ± 4.9 | 28.4 ± 4.9 |
| $1:5^5$ | 11176 | 20.9 ± 3.5 | 35.5 ± 10.5 |
| $1:5^6$ † | 2235 | 25.3 ± 19 | 27.6 ± 33.1 |
| $1:5^7$ | 447 | ND | 49.2 ± 31.6 |
| $1:5^8$ | 89 | ND | ND |
| $1:5^9$ | 18 | ND | ND |
| $1:5^{10}$ | 3,6 | ND | ND |
| $1:5^{11}$ | (0) | ND | ND |

Note

† represents (LOQ) for RT-ddPCR and RT-qPCR, and LOD for RT-qPCR; ND = Not detected.

combination of the negatively charged filter and NucliSENS® lysis buffer performed best in the recovery of *Salmonid alphavirus* from seawater [21].

The MF⁻ filter has been used for concentration of viruses in water, such as polio virus in seawater [35], where an acidification step to remove cations was required to increase the elution of virus from the filter [36–39]. In this study, ISAV was concentrated effectively with the MF⁻/buffer 1 method from natural seawater, without the acidification step, by direct lysis of the adsorbed ISAV with buffer 1. The omission of the laborious acidification step allowed us to maximize virus recovery during the concentration process. ISAV recovery with the MF-/buffer 1 method from natural seawater was higher compared to that recovered from the artificial seawater. In comparing virus recovery between natural and artificial seawater, it is important to note that factors, such as the presence of organic content in the natural seawater, could have an impact on virus recovery. Indeed, several studies have reported the use of membrane filters to recover solid-associated viruses from effluent, raw sewage, and sludge samples [40–42], and it has been suggested that the presence of organic matter in the water enhances virus adsorption to filters.

Environmental water samples are often associated with high levels of inhibitors and low levels of targets, leading to quantification errors by RT-qPCR [43, 44]. In the present study, RT-qPCR and RT-ddPCR were applied in order to check for inhibitors and increase detection sensitivity, respectively. We found inhibition in some of the natural seawater samples using RT-qPCR, which suggests an advantage for the use of the RT-ddPCR assay for these samples [23]. To detect ISAV in seawater, we used published ISAV-specific primers and probe [32]. These primers and probe were specifically optimized for RT-qPCR. Unlike RT-qPCR, where the data are measured from a single amplification curve that is dependent on reaction efficiency, primer dimers, and sample contaminants, RT-ddPCR is measured at the reaction end point, which eliminates these potential pitfalls. Despite these benefits, experimental optimization remains a requirement for RT-ddPCR, to ensure that the final data are interpretable. In order to use the same primers and probe for the RT-ddPCR assay, we first optimized the assay to determine the best conditions for high amplification and good cluster separation of the

positive and negative droplets by varying annealing temperatures and concentrations of primers and probe. By using RNA obtained from the ISAV stock for the RT-ddPCR assay, we chose 60°C as optimal annealing temperature, and primer and probe concentrations of 900 nM and 250 nM, respectively, as the best. Optimisation of these parameters are common for RT-qPCR and RT-ddPCR assays [45, 46]. Based on our experience during the general optimisation of the RT-qPCR processes, small changes in annealing temperature or primer/probe concentration directly influenced the $Cq$ values, and thus the outcome. However, RT-ddPCR was less sensitive to such changes. Changes of 1°C in annealing temperature, or variations of 100 nM in primer/probe concentration, were tolerated according to the optimization results, as shown in Fig 2 and S1 Fig.

During the method development, we defined the optimal limits of quantification and sensitivity of RT-ddPCR and RT-qPCR assays for detection of ISAV from samples concentrated by MF⁻/buffer 1 method, to determine whether the method may be useful for ISAV surveillance in Atlantic salmon populations in the farms. The results show that both RT-qPCR and RT-ddPCR have the potential to be used for the detection of ISAV from samples concentrated by

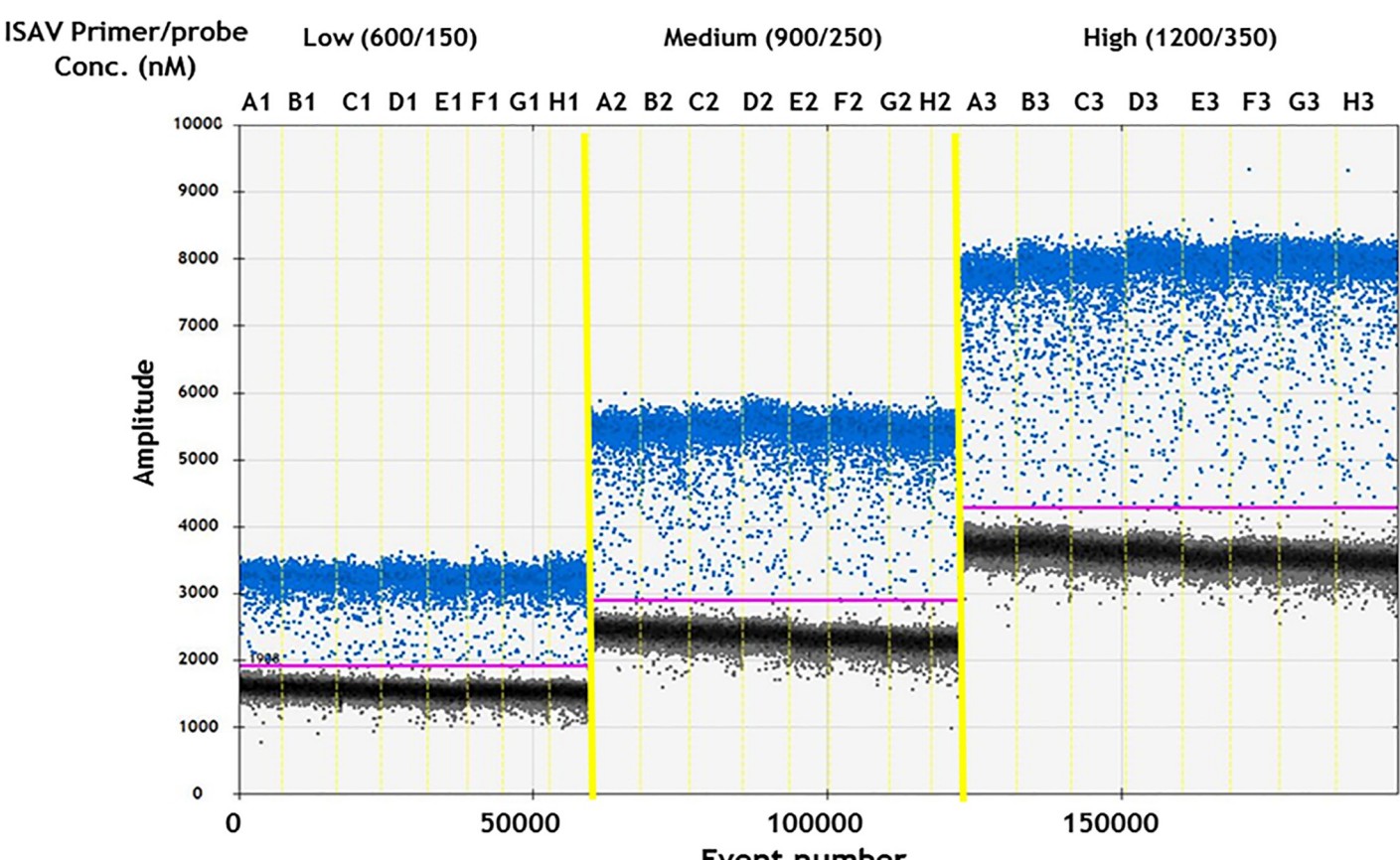

**Fig 2. Optimization of the annealing temperatures and primer/probe concentration for detection of ISAV using RT-ddPCR.** The RT-ddPCR amplitude plot is showing clusters of ISAV negative (black) and positive (blue) droplets. Lanes divided by vertical thin yellow lines (A1 –H1, A2 –H2 and A3 –H3) represent the gradient of annealing temperatures: 63, 62.4, 61.4, 59.9, 58.1, 56.5, 55.6 and 55 ºC. Lanes divided by thick yellow lines represent the gradient of three following primers/probe concentrations: low (600/150), medium (900/250), and high (1200/350 nM). The pink line represents the threshold discriminating droplets.

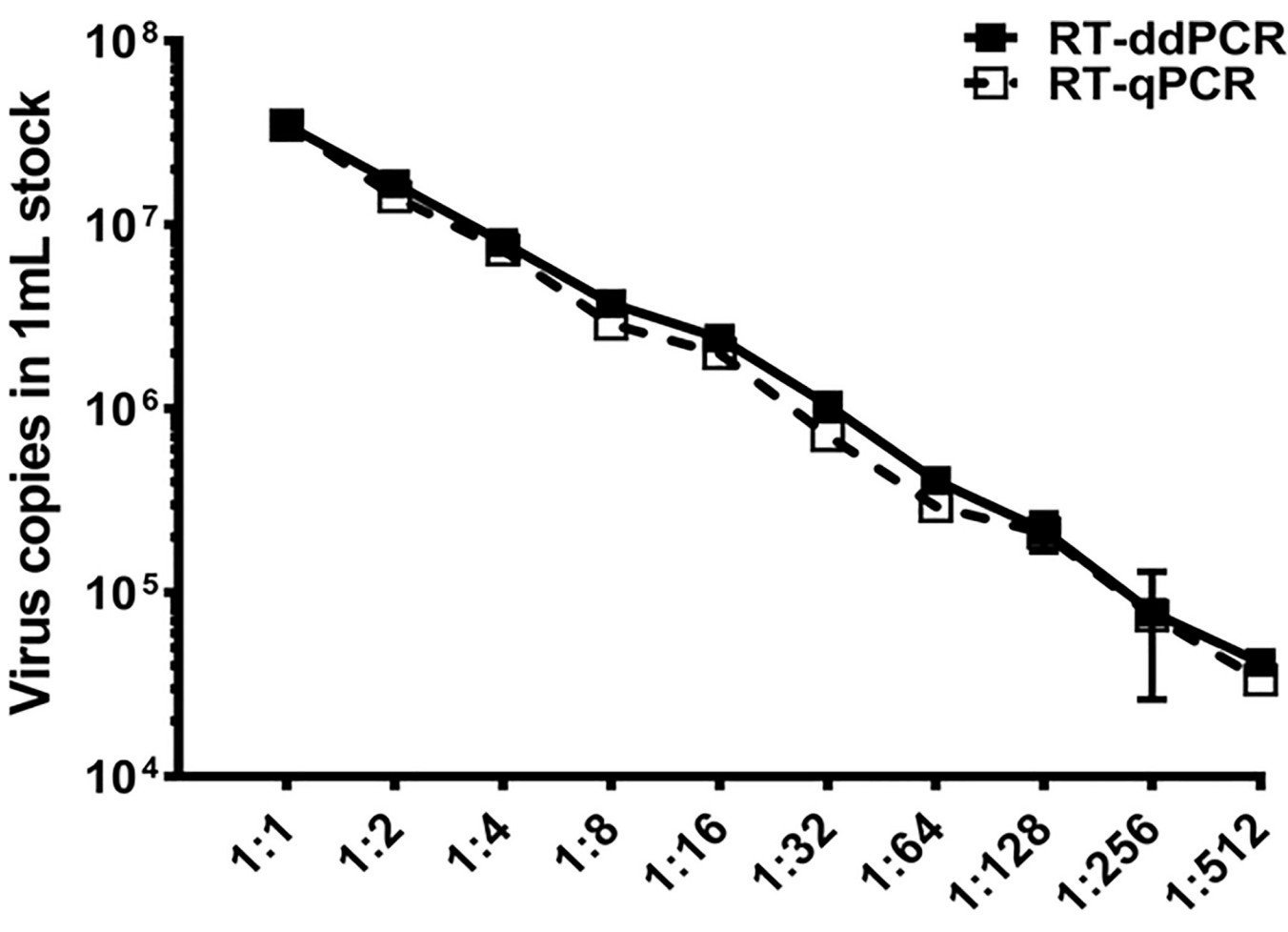

**Fig 3. Quantification of ISAV (copy numbers/ml) in two-fold serial dilutions (1:1–1:2^9) of the stock sample, using RT-qPCR and RT-ddPCR.** The plot represents the assay sensitivity of both PCR methods. Duplicate experiments for both RT-qPCR and RT-ddPCR were performed as technical replicates from the same RNA extracted from the dilution series of the ISAV stock.

the MF⁻/buffer 1 method. Linearity and quantitative correlation with RT-qPCR and RT-ddPCR results were high, as shown in Figs 3 and 4. However, while the sensitivity of both RT-qPCR and RT-ddPCR are somewhat lower ($2.2 \times 10^3$ copies/L), the quantitative accuracy of RT-ddPCR was high, and the assay generated up to 19.2% difference in ISAV recovery relative to RT-qPCR from the natural seawater samples Table 3. Notably, the variable assay differences between RT-qPCR and RT-ddPCR were not observed during the quantification of the ISAV stock Fig 3), which suggests that environmental factors related to the spiked natural seawater sample may have influenced the quantification results. This finding is consistent with our previous study with salmonid alphavirus [47]. However, several studies have also shown divergent results between RT-qPCR and RT-ddPCR assays [48, 49]. In most cases, methods used to determine the concentration of standards, the lack of rigor in the production of standard curves for primer validation, and absolute quantification, have also been attributed to the course of assay variability [50].

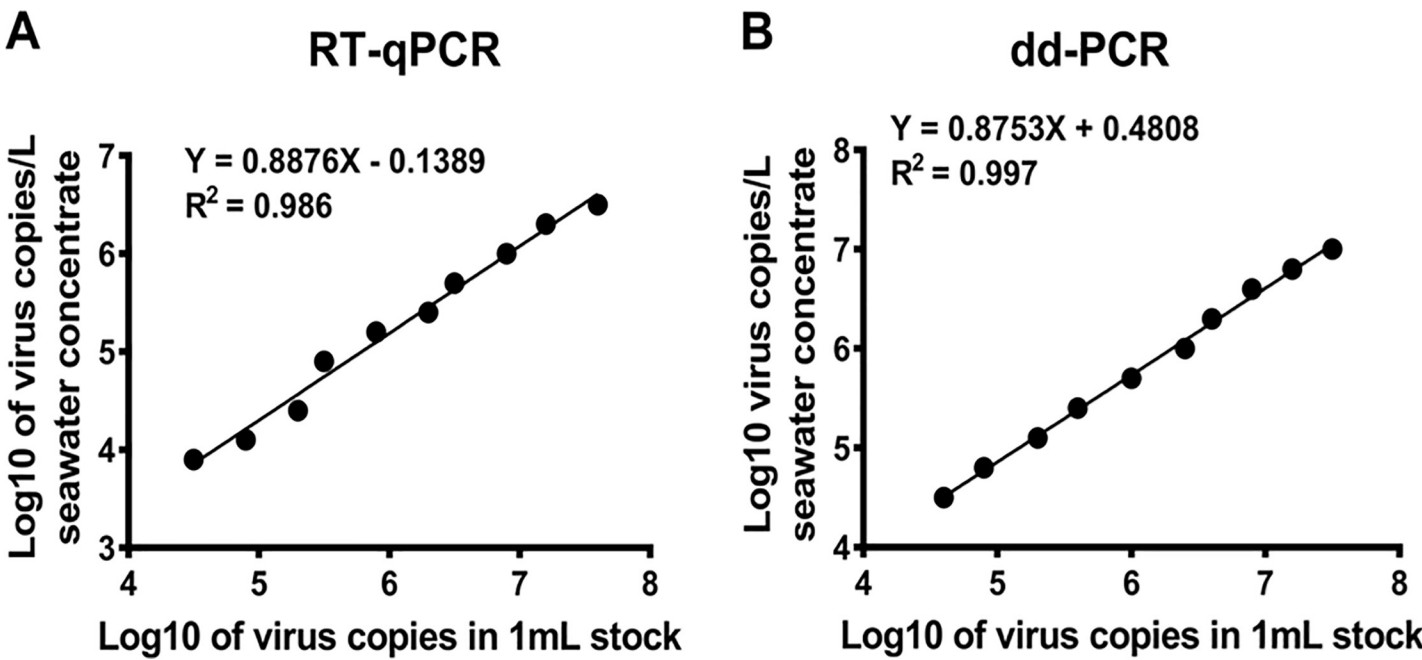

**Fig 4.** Logarithmic regression model for log 10-transformed virus copy numbers in 2-fold serial dilutions of ISAV stock versus log 10-transformed recovered ISAV copy numbers in samples concentrated with the MF⁻/buffer 1 method, and quantified using (A) the RT-qPCR and (B) the RT-ddPCR methods.

Overall, based on the percentages of ISAV recovered from dilution series from natural seawater, measured by RT-qPCR and RT-ddPCR, the negatively charged filter in combination with NucliSENS® lysis buffer as eluent can be used for concentration and detection of ISAV from seawater. The method can potentially co-concentrate other viruses, hence the MF⁻/buffer 1 method of concentration and elution steps described here may be used for the assessment of viruses in seawater samples from tank or pond aquaculture systems, and farmed fish net-pens. The performance of RT-ddPCR and RT-qPCR validates the use of either platform for the detection of ISAV from seawater. However, the variability between RT-ddPCR and RT-qPCR assays suggests trade-offs between assay sensitivity that should be taken into consideration. Also, given the importance that natural seawater was used for the study, and uncertainties that may be associated with RT-qPCR standard curves, the RT-ddPCR performance was more consistent, and thus RT-ddPCR is preferable. However, the high cost of RT-ddPCR reagents and equipment should be taken into consideration when designing experiments or applying the method for disease surveillance under field conditions.

## Supporting information

**S1 Fig. Graph shows ddPCR optimization ISAV recovery (Residual) with the different annealing temperatures and primer/probe concentrations.**
(TIF)

**S1 Table. Show ISAV recovery (Residual) with the different annealing temperatures and primer/probe concentrations.**
(DOCX)

**S1 Data.**
(ZIP)

## Acknowledgments

The authors gratefully acknowledge the generous technical assistance from Marit Leinaas Kjellin, Saima Nasrin Mohammad, and Inger Böckerman.

## Author Contributions

**Conceptualization:** Simon Chioma Weli, Atle Lillehaug.

**Data curation:** Simon Chioma Weli, Haitham Tartor, Bjørn Spilsberg.

**Formal analysis:** Simon Chioma Weli, Haitham Tartor, Bjørn Spilsberg.

**Funding acquisition:** Simon Chioma Weli, Atle Lillehaug.

**Investigation:** Simon Chioma Weli.

**Methodology:** Simon Chioma Weli, Atle Lillehaug.

**Project administration:** Simon Chioma Weli.

**Resources:** Simon Chioma Weli, Atle Lillehaug.

**Software:** Simon Chioma Weli.

**Supervision:** Simon Chioma Weli, Atle Lillehaug.

**Validation:** Simon Chioma Weli.

**Visualization:** Simon Chioma Weli, Haitham Tartor, Bjørn Spilsberg, Ole Bendik Dale, Atle Lillehaug.

**Writing – original draft:** Simon Chioma Weli.

**Writing – review & editing:** Simon Chioma Weli, Haitham Tartor, Bjørn Spilsberg, Ole Bendik Dale, Atle Lillehaug.

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
