## [Decision Letter · Decision Letter 0]

6 Apr 2021

PONE-D-21-01328

Evaluation of Charged Membrane Filters and Buffers for Concentration and Recovery of Infectious Salmon Anaemia Virus in Seawater

PLOS ONE

Dear Dr. Weli,

Thank you for submitting your manuscript to PLOS ONE. After careful consideration, we feel that it has merit but does not fully meet PLOS ONE’s publication criteria as it currently stands. Therefore, we invite you to submit a revised version of the manuscript that addresses the points raised during the review process.

As a comment to the standard text I would like to add that the paper needs minor revision in terms of the scientific content as suggested by reviewer 2. However, as reviewer 1 has commented, a similar paper has already been published with another virus (SAV) by your group. Please consider to shorten the current manuscript accordingly, referring on the SAV paper, stating the modifications of the methods described for SAV recovery from sea water. Actually, I ask myself why both manuscripts weren't combined in one paper.

I am sorry for the delayed reviewing process, but seven other potential reviewers rejected to deal with the manuscript or where uninvited because of missing responses.

We look forward to receiving your revised manuscript.

Kind regards,

Uwe Fischer

Academic Editor

PLOS ONE

Journal Requirements:

Reviewers' comments:

Reviewer's Responses to Questions

**Comments to the Author**

1. Is the manuscript technically sound, and do the data support the conclusions?

Reviewer #1: Yes

Reviewer #2: Yes

2. Has the statistical analysis been performed appropriately and rigorously? 

Reviewer #1: Yes

Reviewer #2: Yes

3. Have the authors made all data underlying the findings in their manuscript fully available?

Reviewer #1: Yes

Reviewer #2: Yes

4. Is the manuscript presented in an intelligible fashion and written in standard English?

Reviewer #1: Yes

Reviewer #2: Yes

5. Review Comments to the Author

Reviewer #1: Evaluation of Charged Membrane Filters and Buffers for Concentration and Recovery of Infectious Salmon Anaemia Virus in Seawater.

The manuscript describes the method for detection and concentration of Infectious salmon anaemia virus in natural and artificial seawater using charged membrane filters. The methods and use of filters including the experimental design and set up is very similar to published work by authors group for salmonid alphavirus as:

Weli, Simon Chioma, et al. "Development and evaluation of a method for concentration and detection of salmonid alphavirus from seawater." Journal of Virological Methods 287 (2021): 113990.

The novel finding in the current manuscript is applicability of previously described method for concentration and recovery of Infectious Salmon Anaemia Virus. The experimental set up is already available in public domain. In my opinion, this work could not be accepted for publication in current form and should be revised highlighting the findings related to ISAV as short communication with reference to published work for methodology.

Reviewer #2: The present study evaluated different membrane filters and buffer for recovery of ISAV from natural and artificial seawater. The authors postulate that routine surveillance of seawater for the presence of ISAV can supplement traditional surveillance programs based on the detection of clinical ISA disease, and provide early warning. The study demonstrated that the negative charged filters combined with NucliSENS® lysis buffer produced the highest ISAV recoveries from natural seawater – and further studies were therefore directed toward the MF-/buffer 1 combination. The authors went on to show that virus recover using either qPCR or ddPCR was generally better at lower viral concentrations. And finally a LOQ was defined for this assay.

In general, this is a well-conceived and excellent written manuscript. Its focus is largely technically based, and addresses an important aspect of eDNA surveillance programs for viruses. I have raised only minor comments/edits that should be considered.

Line 134-137 – The did the researchers include a positive control either directly in the RNA extraction or spiked at the lysis stage? This would help control for loss of RNA during extraction process.

Line 141 – Only 1 ml of lysate was used for the RNA extraction. However, the volume of lysis buffer added to the filters was 4ml therefore only ¼ of the sample was extracted. Was the yield of ISAV adjusted for this? From the equation on line 139 it would appear not.

Line 155 – the authors need to explain what they mean by inhibition and how it was quantified and what was the threshold. Additional info also needs to be provided for the ratios 1:1 and 1:4, was this RNA:seawater in the 2ul template (i.e. 1ul RNA + 1 ul seawater = 1:1). Furthermore, and related to the comment above, if RNA was diluted was the yield calculations adjusted accordingly?

Line 165 – three not tree

Line 213 – what do the authors mean by ‘reliably detected’. Was there a specific threshold used?

Discussion – I would prefer to have seen some discussion putting the results of this study into the context of what we know about ISAV viral shedding. For example, Gregory et al. 2009 (DOI: 10.1111/j.1365-2761.2009.00999.x) has described the shedding rate and minimal infectious dose of ISAV in experimental challenges and reported their results a TCID 50. In the present study the LOQ was defined, but it remains unclear to this reviewer how this compares to the minimum infectious dose and shedding rate. If the LOQ is higher than the minimum infectious dose then the utility of this assay would be limited. The manuscript would be greatly improved if additional discussion concerning these points could be included.

6. PLOS authors have the option to publish the peer review history of their article (what does this mean?). If published, this will include your full peer review and any attached files.

Reviewer #1: **Yes: **Kalpana Agnihotri

Reviewer #2: No

---

## [Author Response · Author response to Decision Letter 0]

18 May 2021

PONE-D-21-01328

Dear Editor,     

Thank you for the positive feedback, and all the constructive comments from you and the reviewers to the manuscript: Evaluation of Charged Membrane Filters and Buffers for Concentration and Recovery of Infectious Salmon Anemia Virus in Seawater 

Authors: Simon Chioma Weli, Haitham Tartor, Bjørn Spilsberg, Ole Bendik Dale, Atle Lillehaug 

We have revised the manuscript as recommend by you and the two reviewers by addressing all the comments raised. We have attached the following files: 

1. A revision showing edits made in this revision as tracked changes (in Word doc format), file name: “Revised Manuscript with Track Changes”.

2. A revision with all changes accepted ('clean' file, in Word doc format), which is an unmarked version of the revised manuscript without tracked changes. File labeled “manuscript”.

3. A response letter, file name: “Response to Reviewers”, which is a rebuttal letter that responds to each point raised by the academic editor and reviewers. 

Editor’s comments: As a comment to the standard text I would like to add that the paper needs minor revision in terms of the scientific content as suggested by reviewer 2. However, as reviewer 1 has commented, a similar paper has already been published with another virus (SAV) by your group. Please consider to shorten the current manuscript accordingly, referring on the SAV paper, stating the modifications of the methods described for SAV recovery from sea water. Actually, I ask myself why both manuscripts weren't combined in one paper.

Author’s response: As suggested by reviewer 1, we have shortened the manuscript and referred to the method as described for SAV recovery from seawater. With regard to why both manuscript (SAV and ISAV) were not combined in one paper. The reason was that, although the entire funding for the research was for developing a method for concentration and recovery of fish viruses (ISAV and SAV) from seawater, the SAV aspects of the research work was specifically organized as part of a PhD thesis. Combining both manuscript (SAV and ISAV) in one paper would be problematic for the student. It will mean that the PhD student would have to defend a subject that is not part of her PhD topic, as published articles derived from PhD thesis work are normally attached and defended during dissertation.

Journal Requirements:

We have revised the manuscript to meet PLOS ONE format and style as requested by the academic editor.

We have reviewed the reference. Change to the reference list: Reference #21 was updated to reflect correct article publication date

Reviewer #1 comment: In my opinion, this work could not be accepted for publication in current form and should be revised highlighting the findings related to ISAV as short communication with reference to published work for methodology.

Author’s response: As suggested by reviewer 1, we have shortened the manuscript (as short communication) without omitting details that will allow readers to have full understanding of the work. We have also referenced the method as described for SAV recovery from seawater. The reduction where mainly focused on the materials and method section. 

Reviewer #2: 

Reviewer comment: Line 134-137 Did the researchers include a positive control either directly in the RNA extraction or spiked at the lysis stage? This would help control for loss of RNA during extraction process.

Author’s response: No, we did not include positive control directly during RNA extraction or spiked at the lysis stage in this particular (in vitro) experiment. However, in our field study, we did add Mengovirus to evaluate extraction efficiency of SAV recovery from seawater. We agree that addition of spike positive control, would have probably helped to check for loss of RNA during extraction process. 

Reviewer comment: Line 141 – Only 1 ml of lysate was used for the RNA extraction. However, the volume of lysis buffer added to the filters was 4ml therefore only ¼ of the sample was extracted. Was the yield of ISAV adjusted for this? From the equation on line 139 it would appear not.

Author’s response: Lines 126 – 130: Yes, the recovery was adjusted. Retrospective (back) calculations was used to calculate ISAV copy number in 4 or 2.4 ml concentrates by taken into consideration, that ISAV was spiked in 1L of seawater, concentrated and eluted in either 4 or 2.4 ml (lysate) and that from the 4 or 2.4ml, RNA was extracted from 1ml of the lysate. The extracted RNA was eluted in 50 µl and 2 µl of RNA was used for PCR. 

Reviewer comment: Line 155: The authors need to explain what they mean by inhibition and how it was quantified and what the threshold was. Additional info also needs to be provided for the ratios 1:1 and 1:4, was this RNA: seawater in the 2ul template (i.e. 1ul RNA + 1 ul seawater = 1:1). Furthermore, and related to the comment above, if RNA was diluted was the yield calculations adjusted accordingly?

Author’s response: Line 145 – 152: We explained inhibition and how it was quantified in the manuscript. Additional, we have also provided detailed information about the dilutions in the manuscript. Environmental samples, such seawater are particularly challenging and usually contain complex mixtures of a variety of inhibitory substances that could impact the RT-qPCR and target quantification. For this reason, RNA was therefore, analyzed undiluted (1:1) and diluted (1:4) in duplicates, by RT-qPCR. Potential inhibition was detected when the Cq difference between the 1:1 and 1:4 samples was found to be less than 2 Cq. For the samples that falls within this threshold, the 1:4 dilution was used to estimate virus quantities. The ratios 1:1 indicate stock RNA (2 µL template) while the 1:4 ratios was stock RNA: RNA elution buffer (i.e. 1ul RNA + 3 µL RNA elution buffer = 1:4). Yes, when 1:4 dilution was used to estimate virus quantities, virus recovery calculations was adjusted accordingly. 

Reviewer comment: Line 165: Three not tree

Author’s response: Line 165: Corrected from tree to three 

Reviewer comment: Line 213: What do the authors mean by ‘reliably detected? Was there a specific threshold used?

Author’s response: Line 213: The statement ‘reliably detected’ has been rephrased to read “detected”, as there was no threshold used 

Reviewer comment: Discussion – I would prefer to have seen some discussion putting the results of this study into the context of what we know about ISAV viral shedding. For example, Gregory et al. 2009 (DOI: 10.1111/j.1365-2761.2009.00999.x) has described the shedding rate and minimal infectious dose of ISAV in experimental challenges and reported their results a TCID 50. In the present study the LOQ was defined, but it remains unclear to this reviewer how this compares to the minimum infectious dose and shedding rate. If the LOQ is higher than the minimum infectious dose then the utility of this assay would be limited. The manuscript would be greatly improved if additional discussion concerning these points could be included.

Author’s response: We agree with the reviewer suggestion; however, the authors deliberately decided to avoid ISAV virus shedding as we have a separate manuscript on ISAV in experimental challenges study where we focused on ISAV viral shedding by comparing virus recovery from seawater and tissues (gills and heart) over a time period.

---

## [Editor Report · Decision Letter 1]

21 May 2021

PONE-D-21-01328R1

Evaluation of Charged Membrane Filters and Buffers for Concentration and Recovery of Infectious Salmon Anaemia Virus in Seawater

PLOS ONE

Dear Dr. Weli,

Thank you for submitting your manuscript to PLOS ONE. After careful consideration, we feel that it has merit but does not fully meet PLOS ONE’s publication criteria as it currently stands. Therefore, we invite you to submit a revised version of the manuscript that addresses the points raised during the review process.

Please see my comments below.

We look forward to receiving your revised manuscript.

Kind regards,

Uwe Fischer

Academic Editor

PLOS ONE

Journal Requirements:

Additional Editor Comments (if provided):

I think you have properly addressed the reviewer's comments. To convert the manuscript into a follow-up short communication is a good compromize.

Please find attached some minor points:

Line 111 and 114: “were used”

Line 144: “copy numbers” “by taking”

Line 166: “such as”

Line 169: “therefore” can be omitted.

Line 171: “that fall”

Line 173: “ratios were”

Line 442: “Graph shows”

Please double-check the manuscript for similar mistakes (e.g. connected with singular/plural mix-ups).

---

## [Author Response · Author response to Decision Letter 1]

27 May 2021

PONE-D-21-01328

Dear Editor,     

Thank you again for the positive feedback, and all the constructive comments to the manuscript: Evaluation of charged membrane filters and buffers for concentration and recovery of infectious salmon anemia virus in seawater 

Authors: Simon Chioma Weli, Haitham Tartor, Bjørn Spilsberg, Ole Bendik Dale, Atle Lillehaug 

We have revised the manuscript as recommend by addressing all the comments you raised. We have attached the following files: 

1. A revision showing edits made in this revision as tracked changes (in Word doc format), file name: “Revised Manuscript with Track Changes”.

2. A revision with all changes accepted ('clean' file, in Word doc format), which is an unmarked version of the revised manuscript without tracked changes. File labeled “manuscript”.

3. A response letter, file name: “Response to Reviewers”, which is a rebuttal letter that responds to each point raised by the academic editor and reviewers. 

Editor’s comments: Please double-check the manuscript for similar mistakes (e.g. connected with singular/plural mix-ups). Please double-check the manuscript for similar mistakes (e.g. connected with singular/plural mix-ups).

Author’s response: As suggested by the editor, we have gone through the manuscript for mistakes. 

In lines 111 and 114: Was has been changed to “were used”

In line 144: We changed copy number to “copy numbers” and taken has been changed to “by taking”

In line 166: We have added such “as”

In Line 169: We have removed “therefore”.

In line 171: We have changed “falls” to fall”

In line 173: We have “ratios were”

In line 442: We change Graph show t0 “Graph shows”

We have also reviewed the reference list to ensure that they are complete and correct.

---

## [Editor Report · Decision Letter 2]

2 Jun 2021

Short Communication: Evaluation of charged membrane filters and buffers for concentration and recovery of infectious salmon anaemia virus in seawater

PONE-D-21-01328R2

Dear Dr. Weli,

We’re pleased to inform you that your manuscript has been judged scientifically suitable for publication and will be formally accepted for publication once it meets all outstanding technical requirements.

Kind regards,

Uwe Fischer

Academic Editor

PLOS ONE

Additional Editor Comments (optional):

Dear authors,

thank you for implementing changes as suggested. Indicated lines for changes do not match with the present document which is probably due to shifts in text length resulting from the respective changes.

I think the manuscript has now reached a quality so that it can be published in PLOS ONE.

With best regards,

Uwe Fischer.
---

## [Editor Report · Acceptance letter]

8 Jun 2021

PONE-D-21-01328R2 

Short Communication: Evaluation of charged membrane filters and buffers for concentration and recovery of infectious salmon anaemia virus in seawater 

Dear Dr. Weli:

I'm pleased to inform you that your manuscript has been deemed suitable for publication in PLOS ONE. Congratulations! Your manuscript is now with our production department. 

Kind regards, 

on behalf of

Dr. Uwe Fischer 

Academic Editor

PLOS ONE